# Analysis of the Impacts of Climate Change on Agriculture in Angola: Systematic Literature Review

**Carlos D. N. Correia** [1,2] [ID], **Malik Amraoui** [2] **and João A. Santos** [2,*] [ID]

1   Huila Polytechnic Institute, Mandume Ya Ndemufayo University, Arimba Main Road, 776, Lubango P.O. Box 201, Angola; al78021@alunos.utad.pt
2   Center for the Research and Technology of Agro-Environmental and Biological Sciences (CITAB), Inov4Agro, University of Trás-os-Montes and Alto Douro (UTAD), Quinta de Prados, 5000-801 Vila Real, Portugal; malik@utad.pt
*   Correspondence: jsantos@utad.pt

**Abstract:** The changing global climate, characterized by rising surface air temperatures, shifting precipitation patterns, and heightened occurrences of extreme weather events, is anticipated to profoundly impact the environment, economy, and society worldwide. This impact is particularly acute in African nations like Angola, where crucial sectors, such as agriculture, rely heavily on climate variability and exhibit limited adaptive capacity. Given that the majority of Angola's agriculture is rain-fed and serves as a vital source of livelihood for the populace, the country is especially vulnerable to climate change, particularly in its southern region. Climate change has caused severe damage in Angola, especially in the southern part of the country, where the worst droughts in decades have affected over 3.81 million people, resulting in food and water shortages. Between 2005 and 2017, climate-related disasters cost the country about 1.2 billion US dollars, further exacerbating the economic and social challenges faced by the population. This study presents a systematic review of the effects of climate change on agriculture in Angola, with a focus on the southern region. Employing the PRISMA2020 methodology, the review examined 431 documents from databases such as Scopus and Web Science, spanning from 1996 to 2023, with 63 meeting inclusion criteria. The review reveals a paucity of research on the short and long-term impacts of climate change on Angolan agriculture. Projections indicate a rise in temperatures and a general decrease in precipitation, with the southern region experiencing a more pronounced decline. Agricultural productivity may suffer significantly, with models suggesting a potential 7% reduction by 2050.

**Keywords:** climatic variables; crops; climate change; temperature and precipitation variability

## 1. Introduction

Widespread and rapid changes in the atmosphere, ocean, cryosphere, and biosphere have occurred. Human-caused climate change is already affecting many weather and climate extremes in every region across the globe. This has led to widespread adverse impacts and related losses and damages to nature and people. Vulnerable communities who have historically contributed the least to current climate change are disproportionately affected [1].

The ramifications of climate change on agriculture are manifold, encompassing shifts in temperature and precipitation patterns, changes in pest and disease dynamics, variations in atmospheric carbon dioxide levels, and modifications in the nutritional quality of crops, among others [2,3].

Particularly in Africa, where agriculture serves as a cornerstone of livelihoods, predominantly rainfed and intricately linked to subsistence, the vulnerabilities to climate change are heightened [4,5]. In Africa, where the specter of food insecurity looms large, climate change amplifies existing challenges, potentially jeopardizing efforts to alleviate

hunger and malnutrition [4]. Southern Africa, heavily reliant on rainfed agriculture, confronts imminent threats to food security and agricultural productivity due to increased rainfall variability and warming temperatures under climate change [6,7]. Angola, marked by vulnerability to climate change and recurrent droughts epitomize the precarious balance between agricultural sustainability and environmental exigencies [8].

The consequences of these shifts extend to staple crops such as maize, sorghum, and millet, impacting their yields and growing seasons [9]. Millet and sorghum, renowned for their resilience to arid conditions emerge as crucial crops in the face of changing climatic conditions, offering potential avenues for adaptation in regions prone to water stress [10,11]. However, the overarching trend of warming poses challenges to the hydrological cycle, altering precipitation patterns and exacerbating drought conditions [12,13].

In light of these considerations, this review delves into the intricate nexus between climate change and agriculture in Angola, exploring the manifold impacts, challenges, and potential avenues for resilience-building in the face of an uncertain climatic future. Through a comprehensive analysis, this study aims to underscore the imperative of proactive measures and policy interventions to safeguard agricultural sustainability and food security amidst the throes of climate change.

*Review Questions*

This study is important because it helps understand the specific threats that Angola faces, enabling the taking of preventive measures and development of adaptation strategies to protect food security, the economy, and the environment. This evidence-based analysis is essential for the sustainable development and well-being of the population of Angola, particularly in the southern region. Therefore, the main research question for this systematic review will be: "What are the impacts of climate change on agriculture in Angola and, particularly the southern region of the country?"

During the study, we sought to answer the following research questions:

Research Question 1: How has climate change affected the precipitation patterns in Angola and what has been the impact of these changes on agricultural growing cycles?

**Hypothesis 1.** *Climate change in Angola has led to significant variability in precipitation patterns, resulting in more frequent droughts and/or intense rains in certain regions. This negatively affected agricultural cultivation cycles, leading to reduced harvests and instability in food production.*

The objective of this question was to understand the impact of climate change on precipitation patterns and agricultural crop cycles in Angola and highlight the importance of adapting to climate change and developing resilient strategies for the agricultural sector.

Research Question 2: How have average temperatures increased over time in Angola and how has this increase affected crops?

**Hypothesis 2.** *Average temperatures have increased in Angola due to climate change, resulting in more extreme heat conditions during the growing season. This has had negative impacts on crops, reducing productivity and increasing the vulnerability of crops to diseases and pests.*

The objective of this question was to find out whether increases in average temperatures have also been observed over the decades in Angola and how they can negatively affect agriculture by reducing the production and quality of crops.

Research Question 3: What are the adaptation strategies adopted by farmers in Angola to deal with the impacts of climate change on agriculture?

**Hypothesis 3.** *Farmers in Angola are implementing adaptation strategies, such as crop diversification, the use of water conservation techniques, and the implementation of precision agriculture systems to mitigate the impacts of climate change.*

The objective of this question is to seek to understand the adaptation strategies implemented by farmers in Angola to deal with the impacts of climate change on agriculture.

## 2. Materials and Methods

Therefore, this study's fundamental objective is to investigate the intricate nexus between climate change and agriculture in Angola in the southern region, exploring the multiple impacts, challenges, and potential paths to building resilience in the face of an uncertain climate future. With this, it is possible to identify and evaluate recent climate changes in Angola, including temperature trends, rainfall patterns, and extreme weather events. Finally, the study aimed to analyze how these climate changes have affected food security and examine relevant climate variables. The review was conducted systematically through books (chapters), peer-reviewed articles, reports, and accredited journals.

The search strategy and inclusion and exclusion criteria for this review were conducted systematically, using the flowchart structure (Figure 1) of the Preferred Reporting Items for Systematic Reviews and Meta-Analysis (PRISMA2020) protocol and consisted of four steps:

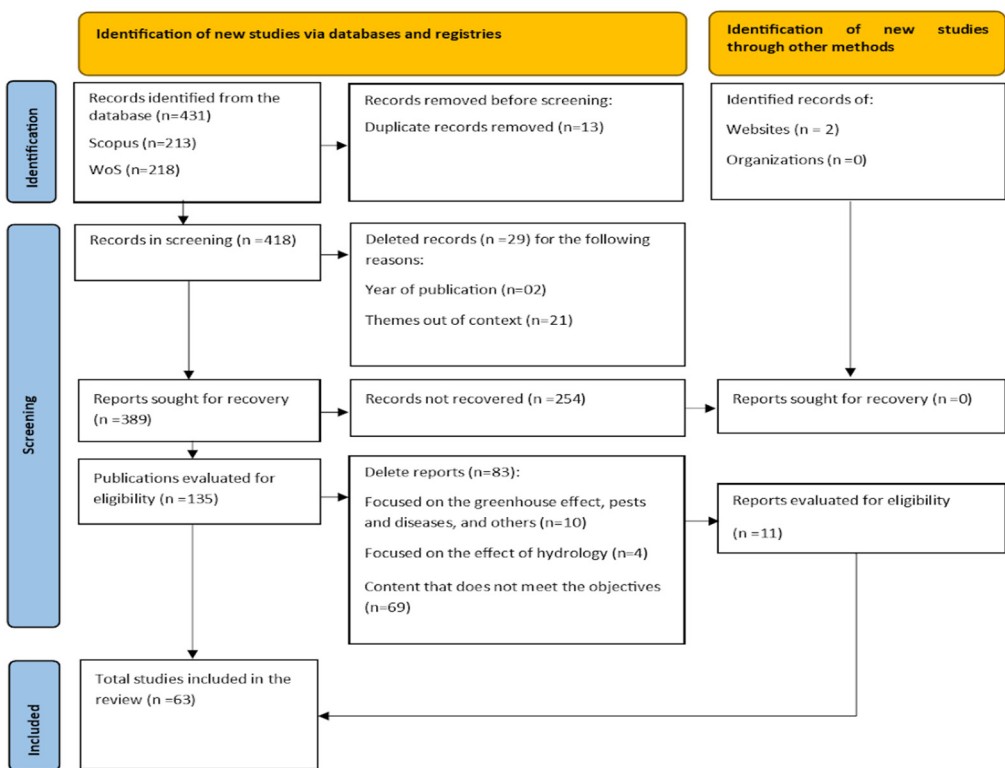

**Figure 1.** Selection process for the systematic literature review guided by the PRISMA 2020 methodology, adapted: systematic review.

Planning how the research would be conducted;

(1)    Defining the databases;

(2)    Pre-selecting articles relevant to the main question, reading their titles, abstracts, and keywords;

(3)    Evaluating the quality of the studies and synthesizing the information.

Search strategy and criteria: the inclusion of the assessment reported below is based on a systematic review of all documents researched through Scopus and Web of Science. A search of the Scopus database resulted in 213 documents that met the search criteria. The Scopus search filter that was followed is: (TITLE-ABSKEY (impact AND climate AND change AND on AND agriculture AND southern AND Africa) AND PUBYEAR > 2000 AND PUBYEAR < 2023.) A search on the Web of Science resulted in 218 articles periodicals and book chapters.

The inclusion criteria followed in this review considered the following: language (documents reported in English, were the only ones considered), the thematic focus (the main topic of the impact of climate change on agriculture in Southern Africa), and the type of document (journals, reports, peer-reviewed articles and chapters of books).

Only documents that met all criteria were considered eligible and included in this review. The initial search was conducted on 19 November 2023, in the Scopus database, and on 15 December 2023, in the Web of Science, resulting in 431 documents published between 1993 to 2023. Before screening, 13 duplicate articles were removed, leaving 418 articles for screening. Of the remaining documents, 29 were excluded after careful analysis (not meeting at least one of the criteria inclusions); 389 documents remained, and of these, 254 were not recovered from the databases. Finally, 135 documents were evaluated for eligibility based on their full text. As a result, 83 documents were excluded after full-text evaluation and review for eligibility. Thus, 52 documents were included (eligible) in the systematic review, as well as 11 publications, using other methods, namely citation searches and websites, totaling 63 eligible documents. The inclusion criteria followed in this study considered language, thematic focus, and type of document.

The search equation varied for each database in ways that allowed the identification of all studies that contained, in the title, the keywords indicated in the search equation above, as shown in Table 1.

**Table 1.** Search equations in databases.

| Database | Search Carried Out | Article Found | Search Date |
|---|---|---|---|
| Scopus | (TITLE-ABS-KEY (impact AND climate AND change AND on AND agriculture AND southern AND Africa) | 213 | 19 November 2023 |
| Web of Science | (TITLE-ABS-KEY (impact AND climate AND change AND on AND agriculture AND southern AND Africa) | 218 | 15 December 2023 |

To achieve the research objectives, some inclusion and exclusion criteria were adopted (see Table 2). It is important to highlight that the inclusion and exclusion criteria were defined based on the main objective of identifying a complete set of documents published since 2000 that are accessible to the general scientific community, focusing on the impacts of climate change on agriculture in Angola, particularly analyzing agroclimatic and bioclimatic indicators. The equation made it possible to identify all studies that contain, in the title, the keywords indicated in the search equations above.

**Table 2.** Criteria for inclusion and exclusion of documents in the systematic search.

| Inclusion Criteria | Exclusion Criteria |
|---|---|
| Were published between 2000 and 2023 | Articles published outside the period 2000 to 2023 |
| Were written in English | |
| Address climate change in agriculture | Works that did not address climate change in agriculture |
| Be an article, thesis, dissertation, or complete report | Articles not available in full and in languages other than English |

## 3. Results and Discussion

### 3.1. Publications Identified with the PRISMA2020 Methodology

The application of the PRISMA2020 methodology made it possible to identify a total of 135 publications through database data and records, as well as 11 publications through other methods, namely citation search and website search (Figure 1). All publications identified in the databases can be accessed at the website address presented in Table 3. The identification of publications through the databases included studies on climate change in agriculture in Africa, with a very small number of studies on climate change in Angola, as well as some regions outside Africa. This motivated the inclusion of these 11 publications

identified by citations and websites searches for articles presenting some useful information about climate change in agriculture in Angola. No article was found that addressed the impact of climate change on agriculture in Angola. One of the relevant publications was on climate change projections from joint multiple models of Regional Experiments Reduction Coordinates (CORDEX) and multi-model sets of Coupled Model Intercomparison Projects (CMIPs) over Angola [14]. Other publications included climate change scenarios for Angola: an analysis of precipitation and temperature projections using four regional climate models (RCMs) [8], drought history and vegetation response in the Angolan highlands [15], and finally, a report from the World Bank Group on Angola, focusing on the country's climate and development [16].

All studies identified in the search were entered into the reference management software Mendeley Desktop and reconfirmed by human verification. In the first stage, 13 publications were excluded as duplicates, which corresponds to 3% of the total (431) publications identified via databases. In the second stage, 29 publications, corresponding to 6.7%, were excluded due to the following: (i) year of publication (0.46%), (ii) topics outside the study context (4.85%), (iii) articles from a systematic review (1.39%), and inaccessible documents (58.93%). In the third stage, 10 articles, corresponding to 2.3%, focused on the greenhouse effect, pests, and pesticides, 4 articles, corresponding to 0.93%, focused on the effect of climate change on hydrology, and finally, 69, corresponding to 16%, whose contents did not align with the objectives of this review, were excluded.

Of the 431 publications initially identified in the databases, citation search, and on the website, 135 remained after removing the number of articles not retrieved from the databases. The 135 documents were published between 1996 and 2023, which corresponds to an approximate average of six publications per year, but 98.7% of these documents were published starting from the year 2000 (Figure 2). A total of 63 documents selected for the literature review were published after 2000, which corresponds to an approximate average of three publications per year (Figure 2). However, 47% of the documents included in the literature review were published in the last 6 years, with 2020 and 2023 being the years in which the largest number of these documents were published (6.9%). This growing trend suggests/demonstrates that this subject has gained interest in recent years within the scientific community. Over the past decade, a growing body of economic research has projected the impacts of climate change on important facets of well-being, such as agriculture, industry, human health, energy demand, and economic growth. Given the natural relationship between climate factors and plant growth, the agricultural sector has been extensively researched.

**Table 3.** Bibliographic databases, website address where search results are saved, number of publications identified (N), access date, and search procedure to identify publications included in the systematic review.

| Database | Website Address | Number | Access Date | Search Equation |
|---|---|---|---|---|
| Scopus | https://www.scopus.com/ accessed on 11 March 2024 | 213 | 19 November2023 | Search equation |
| WoS | https://www.webofscience.com/ accessed on 11 March 2024 | 218 | 15 December2023 | Search equation |
| Academic Google | https://www.sciencedirect.com/ accessed on 11 March 2024 | 1 | 30 January 2024 | Citation search |
| | https://www.sciencedirect.com/ accessed on 11 March 2024 | 1 | 30 January 2024 | Citation search |
| | https://iopscience.iop.org/ accessed on 11 March 2024 | 1 | 30 January 2024 | Citation search |
| | https://rmets.onlinelibrary.wiley.com/ accessed on 11 March 2024 | 1 | 30 January 2024 | Citation search |
| | http://iopscience.iop.org/ accessed on 11 March 2024 | 1 | 30 January 2024 | Citation search |
| | https://scholar.google.com.br/ accessed on 11 March 2024 | 1 | 30 January 2024 | Citation search |
| | https://royalsocietypublishing.org accessed on 11 March 2024 | 1 | 30 January 2024 | Citation search |
| | https://www.tandfonline.com/ accessed on 11 March 2024 | 1 | 30 January 2024 | Citation search |
| | https://digitalcommons.unl.edu/ accessed on 11 March 2024 | 1 | 30 January 2024 | Citation search |
| | https://iopscience.iop.org/ accessed on 11 March 2024 | 1 | 3 February 2024 | Website |
| | https://documents1.worldbank.org/ accessed on 11 March 2024 | 1 | 30 January 2024 | Website |

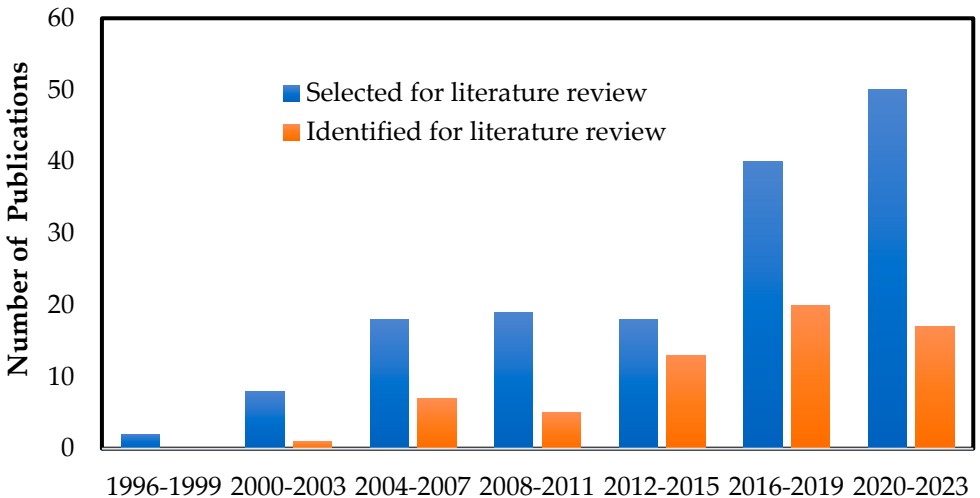

**Figure 2.** Annual numbers of documents identified by literature search (1996–2023) and selected/included for the literature review.

Of the articles selected for review, on the topic in question, only five documents discuss Angola, corresponding to 7.9%, and none of them address the impact of climate change on agriculture. Of the 22 articles related to Southern Africa, only one makes mention of Angola; that is, the majority of documents report/describe studies carried out in other Southern African countries. Some of the studies deal with Africa in general, others address

specific regions of Africa, while other studies are related to South Africa, Zimbabwe, and Mali (Figure 3).

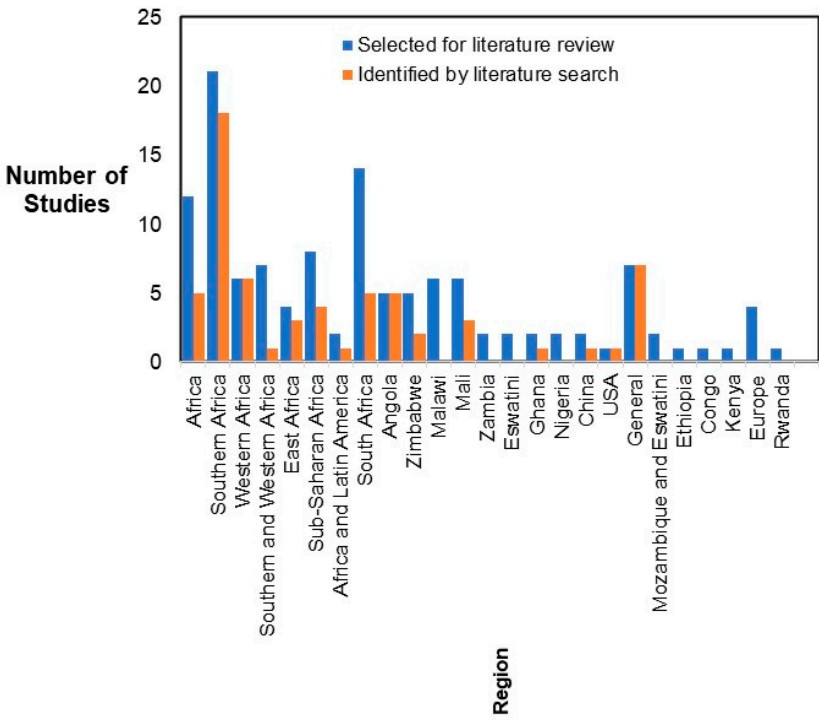

**Figure 3.** Number of publications related to each country in the study region (Southern Africa) identified by the literature search and selected/included for the literature review (1996–2023).

An analysis of the study areas of the 63 documents selected for the review, including records, citation searches, and websites (Figure 3), reveals that the region where the most studies were carried out was Southern Africa with 18 (28.57%). Additionally, studies were conducted in other regions, such as Africa, five (7.9%); West Africa, six (9.52%); Southern and West Africa, one (1.58%); East Africa, three (4.76%); Sub-Saharan Africa, four (6.34%); Africa and Latin America, one (1.58%); South Africa, five (7.9%); Zimbabwe, two (3.17%); Mali, three (4.76%); Ghana, one (1.58%); China, one (1.58%); USA, one (1.58%); Angola, five (7.9%); and finally, seven (11.11%) generally address climate change in agriculture without specifying the region.

These results reveal the insufficiency of research on climate change in agriculture in Angola and the need to acquire and/or update knowledge on this topic. This need is particularly evident and urgent for the southern region of the de country, for which there is a greater lack of comprehensive and accurate information about this natural disaster. Furthermore, 30% of these documents include studies focused on adaptation, which is a key factor that will determine the future impacts of climate change on crop yields, food production, and the adaptive capacity of farmers and their communities. Adaptation is widely recognized as a vital component of the response to climate change. Without adaptation, climate change will hit the sector hard.

### 3.2. Literature Review

In this section, the statistics presented are related to the 63 articles, book chapters, and reports selected for the literature review. This section of the article discusses in a little more detail the impacts of climate change on agriculture in Angola, with a particular focus on the southern region. Several independent studies using a variety of climate models and emissions scenarios indicate that climate change will have an overall negative

impact on African agriculture. Some of the key assessments on African agriculture are discussed below.

### 3.2.1. Climate Change in Agriculture

Although overall agricultural productivity has increased, climate change has slowed this growth over the past 50 years globally, with related negative impacts mainly in mid- and low-latitude regions, but positive impacts are evident in some high latitude regions [1].

Extreme weather events, such as heat stress, frequent droughts, floods, rising temperatures, and pest invasions are all manifestations of climate change that have negative effects on Southern African agriculture [17,18].

Of the articles included in the systematic review, 20.63% (13/63) address Africa's vulnerability to climate change. Africa is believed to be the region most vulnerable to the impacts of climate variability and change. Agriculture plays a dominant role in supporting rural livelihoods and economic growth in most of Africa [19,20]. The problem of vulnerability to climate change is particularly acute in Southern Africa. This region is characterized by a low density of observations and is highly dependent on rural agriculture, where the impact of precipitation changes on maize cultivation depends critically on the timing of the phenological cycle of the crops [21].

Among these articles, 30.15% (19/63) reported that Southern Africa is vulnerable to the effects of climate change due to its rainfed agriculture [2,6,18,20,21]. These findings underscore the region's exposure to different climate shocks and its vulnerability. Of the five articles identified that address Angola, 80% (4/5) state that according to the Notre Dame Global Adaptation Index (ND-GAIN)—an index that illustrates the relative resilience of countries compared to each other—Angola ranks as the 49th most vulnerable country and the 15th least prepared country to deal with the impacts of climate change and other stressors [14,16]. Angola has been characterized as one of the countries most vulnerable to climate change, more specifically in the south, which suffered periods of drought from 2012 to 2022 and is dependent on rainfed agriculture [15,16].

### 3.2.2. Impact of Climate Change on Agriculture

From the literature review, six main factors for the impacts of climate change on agriculture were identified:

(i)     Drought, floods, and food security [2,6,15,18,22–32];
(ii)    Temperature increases, changes in precipitation patterns [12,14,18,20,24,25,27,33,34];
(iii)   Extreme weather events [7,24,35];
(iv)    Occurrence of pests and diseases in plantations [2,6,35];
(v)     Water and energy insecurity [2,36].

### 3.2.3. Impact of Variable Climate on Crop Yields

Half of the documents included in the systematic review, corresponding to 50.79% (32/63), state that temperatures and precipitation are important climate variables used to determine the impacts of climate change at different scales. These two climate variables have a significant effect on crops and their yield. Although rainfall affects agricultural production in terms of photosynthesis and leaf area, temperature affects the length of the growing season [7,10,20,22–25,27–32,34,36–45]. It is important to highlight other indicators, which were found in some articles, such as (i) drought 7.9% (5/63), (ii) humidity 1.5% (1/63), (iii) evaporation 1.5% (1/63) [12], as consequences of temperature and precipitation. Meteorological drought is characterized by a deficit of precipitation in relation to the normal value, characterized by a lack of water induced by the imbalance between precipitation and evaporation, which in turn depends on other elements, such as humidity [25,36]. Temperature and precipitation are the most important variables in determining the impact of crops. Rising temperatures and decreasing precipitation patterns harm crop yields. Crops, especially corn, require more water. This leads to a reduced food supply, as well as overall food availability [46,47].

Maize is the staple food crop in Southern Africa. The main cereals consumed in the Southern African Development Community (SADC) region include millet, rice, sorghum, barley, wheat, and corn. Maize production is declining in most countries in the region, mainly due to extreme droughts and floods [31,36,46,47]. Cereals occupy more than 50% of the region's agricultural land, with corn representing more than 40% of the total harvested area [18].

One of the most studied crops in terms of temperature response is corn, for which increasing temperature shortens the life cycle and the duration of the reproductive phase, causing a reduction in grain yield [42]. On the other hand, drought impacts agriculture by increasing (i) evaporation, atmospheric moisture retention capacity, and water scarcity [42,48], as well as (ii) food and energy security [23,31,48].

A total of 60% (3/5) of the documents that discuss Angola, included in the review, recognize that climate change is not just a future threat, but already a reality in Angola. A state-of-the-art climate impact assessment confirmed that warming has accelerated significantly in recent years. The average annual temperature has increased by 1.4 °C since 1951 and is expected to continue to increase. Precipitation trends are more uncertain, but rainfall variability is increasing, with longer dry spells, worse droughts, and also more floods [14,15,33]. In 2002/2003, an estimated 13 million people faced food shortages as a result of severe droughts in Southern Africa [26].

In Southern Africa, during the 2015/2016 season, there were widespread crop losses due to the occurrence of hot days, heat waves, and heavy rainfall during the rainy season. As stated above, drought tends to involve both a lack of rain and high temperatures, and it is the combined effect of large anomalies in these variables that damage agriculture.

The drought induced by the 2015/16 El Niño Southern Oscillation (ENSO) affected the entire region, resulting in more than 40 million people becoming food insecure and requiring international assistance. The drought caused 643,000 livestock deaths and an overall deficit of corn (the region's staple crop) of 5.1 million tones, representing a 10% decrease in production compared to the previous year and a 15% drop compared with the average of 5 years. The particular case of Angola is shown in Table 4 [2,36].

**Table 4.** Maize production deficit resulting from the drought during 2015/2016 and the number of people affected, adapted [49].

| Country 2011–2016 Average | 2015 Corn Production | 2016 Corn Production | % Change 2015/2016 | Number of People Affected |
|---|---|---|---|---|
| Angola 1366 | 878 | 1500 | −20 | 756,000 |

In 2019, the Angola Vulnerability Assessment Committee mentioned that drought caused livestock loss and low agricultural yields in the southern provinces of Angola. It was estimated that 1.14 million people suffered from food insecurity in the southern provinces of Namibe, Cunene, Cuando Cubango, and Huíla alone, and subsistence communities in the Angolan highlands likely experienced the same hardships [15].

Angola also suffered droughts during 2020/2021, when accumulated rainfall was 30% below the long-term average (the Food and Agriculture Organization of the United Nations). The Angolan people are vulnerable during prolonged periods of drought: a total of 11.1 million people (37%) of the population live in rural regions, and the majority of them practice rain-fed subsistence agriculture [15]. The impacts of climate change also take a heavy toll: climate-related disasters (floods, storms, droughts) cost Angola around 1.2 billion US dollars between 2005 and 2017, and, on average, droughts alone affect around one million Angolans every year [16].

Southern Angola has been the hardest hit and has experienced severe and prolonged droughts over the past decade, with conditions described as the worst in 40 years. In 2021, about 3.81 million people in the six southern provinces had insufficient food, and more than

1.2 million people continue to face water shortages because of droughts [15,16]. Table 5 shows the succession periods of droughts in Angola.

**Table 5.** Common dry periods in Angola from 1981 to 2020, according to each SPI calculation and ENSO years [15].

| Year | Starting Months and End of the Drought | Duration (Months) | Dry throughout Year/Season Rainy/Dry | Max SPI Rating SPI Overlay (Max = 4) | ENSO (Year) Strength TS Anomaly (NOAA, 2021, 2022) |
|---|---|---|---|---|---|
| 1981–1982, | Oct-1981–Jan-1982 | 4 | Wet | Serious, 3 | Neutral (1980–1982) |
| 1989–1990, | Oct-1989–May-1990 | 8 | Wet | Extreme, 3 | Neutral (1989–1990) |
| 1994 | Mar-1994–Oct-1994 | 8 | Wet + Dry | Extreme, 3 | El Niño (1994–1995), moderate |
| 1995–1997 | April-1995–September-1997 | 33 | Many years | Extreme, 4 | La Niña (1995–1996), moderate. Neutral (1996–1997). El Niño (1997–1998), very strong |
| 1999–2001 | Nov-1999–Oct-2001 | 24 | Many years | Extreme, 4 | La Niña (1999–2000), strong. La Nina (2000–2001), weak |
| 2014–2016 | Nov-1999–Oct-2001 | 17 | All year round | Extreme, 4 | El Niño (2014–2015), weak. El Niño (2015–2016), very strong |
| 2017–2018 | Jan-2017–Feb-2018 | 14 | All year round | Extreme, 4 | Niño (2014–2015), weak. El Niño, very strong |
| 2018–2020 | Oct-2018–Feb-2020 | 17 | All year round | Extreme, 4 | La Niña (2017–2018), weak El Niño (2018–2019), weak. Neutral (2019–2020) |

### 3.2.4. Relationship between Climate Variables and Agricultural Crops

Of the documents identified that address Angola, 100% (5/5) state that in Angola, climate change is already being observed. Increasing trends of extreme heat and extreme precipitation events have been found in many regions of Angola, accompanied by a general decrease in annual rainfall. There has also been a notable increase in the frequency of droughts in several regions, and this is expected to persist in the future, even if global warming is stabilized [36,43,49–51]. There is a decreasing trend of precipitation in a northeast-southwest direction over Angola. The lowest precipitation values (<400 mm) occurred in the south (southwest) [8].

Agricultural species respond differently to temperature and precipitation throughout their life cycles. Each species has a defined range of maximum and minimum temperatures within which growth occurs and an optimal temperature at which plant growth progresses at its fastest rate [52,53].

Of the 50.79% (32/63) articles stating that temperature and precipitation are important climate variables used in determining the impacts of climate change at different scales and with a significant effect on crops and their yield. Only 4.76% (37/63) showed a direct relationship between agroclimatic and bioclimatic indicators with crops, as shown in Table 6.

**Table 6.** The direct relationship between agroclimatic and bioclimatic indicators with crops.

| Agroclimatic Variable | Author(s) | Culture | Consequences |
|---|---|---|---|
| Exposure to temperatures above 35 °C | [35,42,54] | Corn | Pollen viability decreases, which negatively impacts grain set, resulting in reduced yields. Thermal stress in crops and water loss through evaporation. |
| Daytime temperatures > 35 °C | [54] | Cereals | They limit the growth and overall productivity of cereals. |
| Extreme daytime temperatures | [54] | Cereals | It causes rapid depletion of soil moisture through an increase in evapotranspiration. |
| Increase in minimum temperatures | [54] | Arvenses | Potentially promotes early senescence, which in turn shortens the grain-filling period, resulting in low yields. |
| Lack of precipitation | [4] | Cereals | Extend growing seasons. |
| Heavy precipitation | [4] | Cereals | They cause soil flooding, anaerobicity, and reduced plant growth. |

*3.3. Food Safety*

Food security is the condition under which all people, at all times, have physical, social, and economic access to sufficient, safe, and nutritious food that meets their dietary needs and food preferences for an active and healthy life [18,32].

Food security is closely affected by climate change, especially by the frequent occurrence of extreme weather events, such as heat waves, droughts, declining soil fertility, pest infections, plant disease, and land degradation [55]. Precipitation and soil moisture are necessary conditions for food production, and irregular rainfall leads to moisture insecurity for agricultural production, which in turn leads to reduced food production. On the other hand, the rising level of $CO_2$ has led to a gradual increase in global temperature, which shortens the long agricultural production cycle, thus causing incalculable damage to food production [56].

Of the documents included in the literature review, 22.22% (14/63) describe studies on the impacts of climate change on agriculture in Angola. The analyses of climate variables concluded that there are direct consequences of global climate change on crop yields, and, therefore, on food security [10,15,18,23,27,31,32,47,57–60].

However, it is a huge task for Southern Africa to achieve food security, given the multi-stressor environment in which agriculture is practiced in this region. It is well-recognized that climate change largely contributes to poor agricultural productivity and food insecurity in Southern Africa and other developing nations.

**4. Climate Change Projections in Angola**

Of the five studies that discuss Angola, three state that the predictions of increased temperature and variability in precipitation are becoming a reality. Regarding the analysis of projections of temperature and precipitation changes in Angola during the 21st century, based on four regional climate models (RCMs), relevant changes in temperature and precipitation indices are projected for the different models in both emission scenarios.

These changes include an increase in both maximum and minimum temperatures of up to 4.9 °C at the end of the century and an intensification of droughts. Precipitation projections are highly variable—rising and falling region-wide, dependent on RCMs. Despite these differences, precipitation generally decreases over time (approximately 2% by 2100), with the southern region of Angola experiencing a stronger drop in precipitation [8].

Climate change projections based on a multi-model ensemble of CORDEX and 4CMIPs over Angola predict an increase in temperature and a decrease in precipitation for two time periods: 2020–2040 and 2040–2060 [14].

The conclusions from the observed data set indicate that the average annual temperature in Angola has increased by an average of 1.4 °C since 1951, with a warming rate of approximately 0.2 [0.14–0.25] °C per decade. However, the precipitation pattern appears to be influenced mainly by natural factors. Therefore, in terms of precipitation, projections

suggest a changes in the rainy season, with an increase in extreme events, such as droughts that could change in all river basins in Angola, and we found growing uncertainty about droughts in the future. Projections show an increase in extreme temperatures, with cooler nights projected to become warmer and days hotter [14].

The Climate and Development Report of the Country of Angola (CCDR) predicted a not very promising future: climate models predict an increase in temperatures, with most of Angola expected to become 1–1.5 °C warmer in 2020–2040 compared to the period 1981–2010, with an increase of 1.4 °C in the average annual temperature already recorded. The imperative to adapt and transition to a proactive climate risk management model is urgent [16].

Agriculture will be hit hard, and the model shows that agricultural productivity would be up to 7% lower by 2050 compared to a scenario without climate damage. Many Angolans who are vulnerable to falling into poverty live in areas of high exposure to climate change, which will make it more difficult for the country to achieve its poverty reduction goals [16].

For Angola, staple crops such as corn, beans, and peanuts, as well as all crops important for food security and nutrition, will be the crops most negatively impacted by climate change [16]. Cassava, millet, sorghum, and bananas, which are crops more tolerant to drought, will be less impacted, and more climate-resilient crops such as cassava will see an increase in suitability, especially in the central regions. Southern regions will generally become less suitable for agricultural production [16].

## 5. Climate Change Adaptation Strategies for Angola

Adapting agriculture to climate change must be a priority to improve crop performance and resilience to environmental constraints. The predicted climate change scenarios, leading to higher temperatures and a water deficit, will impose severe limitations on crop yields and cultivated areas [36,61].

The Angolan people are vulnerable during prolonged periods of drought: a total of 11.1 million people (37% of the population) live in rural regions, and the majority of them practice rain-dependent subsistence agriculture [15]. Therefore, there is a need to adopt adaptation measures to mitigate the impacts of climate change on agriculture.

For the southern region specifically, the adaptation strategies suggested for crops and cultivation systems include the genetic improvement of corn and the development of new cultivars with high productive potential and better adaptation to water deficit conditions and high temperatures. These include millet (*Pennisetum glaucum*) and sorghum (*Sorghum bicolor*), and in the case of flooding, rice is the only main cereal adapted to flooding.

As for the cultivation system, we suggest adjusting the cultivation calendar to avoid the hottest periods of the year in order to help reduce thermal stress on plants.

Regarding the agricultural system, it is suggested to invest in sustainable irrigation systems, such as drip systems and rainwater capture, to guarantee the supply of water to crops during periods of drought. Additionally, introducing sustainable agricultural techniques, such as conservation and the use of cover crops, can improve soil health and increase crop resilience to extreme weather conditions.

Finally, providing training and education to farmers on sustainable agricultural practices, water management, and climate change adaptation techniques, and having early warning systems in place to help communities prepare for extreme weather events such as droughts and floods, can reduce the damage caused by these events.

## 6. Study Limitations and Final Considerations

It is important to mention the limitations of the results of the literature review, as these results depend on the documents selected and analyzed. The research equation was defined in the most generic way possible to achieve the main objective, which was to identify the state-of-the-art on the impact of climate change on agriculture in Angola.

The inclusion of words, Southern Africa, in the search equation results from the insufficient number of articles found in the databases when we searched for the term

Angola. In the first search, using Angola, only five articles were available in the databases; therefore, to increase the number of articles for our systematic review, we included Southern Africa, since Angola is a country that is part of Southern Africa [36].

The insufficiency/lack of papers that address the impacts of climate change on agriculture in Angola influenced the fact that some questions raised in the introduction to our review were not answered. To cite a few examples: what was left unanswered was whether farmers in Angola are implementing adaptation strategies, such as crop diversification, the use of water conservation techniques, and the implementation of precision agriculture systems to mitigate the impacts of climate change. Our study cited a significant number of articles, and most of these articles were used in the introduction to present/describe fundamental concepts and definitions of the study.

As can be seen, there is little research on the impacts of climate change on agriculture in Angola, and this greatly influenced the preparation of this article. For example, of the five articles mentioned here that discuss Angola, only one was found in the Web of Science database [15], two by citation searches [14,51], and two on websites [14,16]. Therefore, there is a need to allocate financial resources for research related to underexplored areas of climate change and to raise awareness about the importance of underexplored areas for understanding climate change.

Studying climate change and its impacts in many African countries is particularly difficult due to the lack of infrastructure and funds needed to collect data and conduct studies. One such country is Angola, where, after 1974, the quantity and quality of meteorological records were considerably reduced [8].

## 7. Conclusions

All of the documents identified that address Angola, 100% (5/5) state that in Angola, climate change is not just a future threat, but already a reality. Similarly, across the world, the average annual temperature has increased by 1.4 °C since 1951 and is expected to continue to increase. Precipitation trends are more uncertain, with longer dry spells, worse droughts, and also more floods. In conclusion, projections point to a significant impact of climate change on Angola's agriculture, especially affecting crops vital for food and nutritional security, such as corn, beans, and peanuts.

The Angolan people are vulnerable during prolonged periods of drought: a total of 11.1 million people (37%) of the population live in rural regions and most of them practice agriculture and rain-dependent livelihood.

Southern Angola has been hardest hit and has experienced severe and prolonged drought over the past decade, with conditions described as the worst in 40 years. In 2021, about 3.81 million people in the six southern provinces had insufficient food, and more than 1.2 million people faced water shortages because of drought.

These climatic changes are anticipated to have profound implications for agriculture in Angola. Models suggest that agricultural productivity could decrease by up to 7% by 2050 due to climate change alone.

In conclusion, this systematic review highlights the urgent need for enhanced research and adaptive strategies to address the impacts of climate change on agriculture in Angola, particularly in the vulnerable southern region. By understanding the specific challenges and developing targeted interventions, policymakers and stakeholders can work towards building resilience and ensuring food security in the face of a changing climate.

**Funding:** This research was funded by National Funds by FCT—Portuguese Foundation for Science and Technology, under the project UIDB/04033/2020 and LA/P/0126/2020 (https://doi.org/10.54499/UIDB/04033/2020).

**Conflicts of Interest:** The authors declare no conflict of interest.

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
