# Peer review of "Analysis of the Impacts of Climate Change on Agriculture in Angola: Systematic Literature Review"

_agronomy, doi:10.3390/agronomy14040783_

Round 1

Reviewer 1 Report

Comments and Suggestions for Authors

This work is too superficial with too little synthesis of the changes that may occur in Angola agriculture to create an in-depth understanding. I find that it is missing important details that are necessary to thoroughly provide a good understanding. Additionally, the manuscript frequently the same effects of climate change and does not discuss in-depth.

There are additional issues with the quality of the writing and figures that need to be addressed before this manuscript could be considered for publication.

Below, please find the comments:

The abstract begins by addressing vulnerability in Angola and in the next sentence mentions the global impact. We suggest an organization of the idea, as it would be more interesting to start from the global and then narrow it down to the more specific, in this case Angola.

Keywords repeat words contained in the title

Line 22: What were the criteria?

Introduction: The research paper introduction aims to present the topic to the reader. It needs to be brief, captivating, and well-referenced. The introduction serves the purpose of leading the reader from a general subject area to a particular field of research. It establishes the context of the research being conducted by summarizing current understanding and background information about the topic, stating the purpose of the work in the form of the hypothesis, question, or research problem, briefly explaining your rationale, highlighting the potential outcomes your study can reveal, and describing the remaining structure of the paper.

In this paper, the introduction is very long, tiring, with repeated information throughout the text, unconnected sentences, and does not highlight the importance of the work.

Need for updated information on climate scenarios

Recent IPCC reports

References 2 and 3 do not represent the subject

Line 46-50: Repeated words, the sentence will need to be rewritten. Another important point: the increase in the concentration of carbon dioxide is the cause of climate change.... it is necessary to review the way of entering information...

Line 51-53: “Among others [5,6]. It is known that agriculture is  an economic activity highly dependent on weather and climate to produce the food  and fibre necessary to sustain 34 human life [7,8] “: incomplete sentence, with numbering in the middle, requires revision.

Line53-57: Unnecessary information

Page 5 - 6: The way it was written is not appropriate for a review article

Line 201-206: This methodology is already known worldwide, there is no need to explain the method. What you need to do is show how the Prisma method was used for this research

Line 251: repeated information

Author Response

This work is too superficial with too little synthesis of the changes that may occur in Angola agriculture to create an in-depth understanding. I find that it is missing important details that are necessary to thoroughly provide a good understanding. Additionally, the manuscript frequently the same effects of climate change and does not discuss in-depth.

There are additional issues with the quality of the writing and figures that need to be addressed before this manuscript could be considered for publication.

Response: The authors are grateful to the reviewer for the pertinent suggestions and comments, as well as for the time spent on this revision. Please find our point-by-point replies below (in red font), along with the corresponding changes in the manuscript (also in red font).

We understand your concern about the depth of our article. When carrying out the research for this work, we came across a limited availability of academic literature that directly deals with the subject in that region, which is why the search equation includes Southern Africa to cover the study, as Angola is part of this African region. We made every effort to address the topic comprehensively, using all available bibliographic sources. We recognize that the desired level of depth may not have been fully achieved due to the scarcity of available material. Therefore, we have made an effort to deepen the study, reorganizing parts of the text and adding more references. As for additional issues relating to the quality of writing and figures, we will make a point of correcting them in ways that improve the work.

Comment 1: The abstract begins by addressing vulnerability in Angola and the next sentence mentions the global impact. We suggest an organization of the idea, as it would be more interesting to start from the global and then narrow it down to the more specific, in this case Angola.

Response 1: We agree on the organization of the summary. We made a point of structuring it in ways that are perceptible and leave a clear and concise idea of what was carried out.

Line 12-28: “The changing global climate, characterized by rising surface air temperatures, shifting precipitation patterns, and heightened occurrences of extreme weather events, is anticipated to profoundly impact the environment, economy, and society worldwide. This impact is particularly acute in African nations like Angola, where crucial sectors such as agriculture heavily rely on climate variability and exhibit limited adaptive capacity. Given that the majority of Angola's agriculture is rain-fed and serves as a vital source of livelihood for the populace, the country is especially vulnerable to climate change, particularly in its southern region. This study presents a systematic review of the effects of climate change on agriculture in Angola, with a focus on the southern region. Employing the PRISMA2020 methodology, the review examined 431 documents from databases such as Scopus and Web Science, spanning from 1996 to 2023, with 63 meeting inclusion criteria. The review reveals a paucity of research on the short and long-term impacts of climate change on Angolan agriculture. Projections indicate a rise in temperatures and a general decrease in precipitation, with the southern region experiencing a more pronounced decline. Agricultural productivity may suffer significantly, with models suggesting a potential 7% reduction by 2050. These findings underscore the urgent need for enhanced research efforts and adaptive strategies to mitigate the adverse effects of climate change on Angolan agriculture, safeguarding both food security and livelihoods.”

Comment 2: Keywords repeat words contained in the title

Response 2: We agree with the suggestion. As such, we have revised the keywords to avoid repetition with the title. We replaced the keywords, as follows: “climatic variables; crops; climate change; temperature, precipitation”.

Comment 3: Line 22: What were the criteria?

Response 3: The authors thank you for the question. In Table 2 the search equation for the selection of the publications is provided, as well as the number of articles found in the databases. Furthermore, Table 3 shows the inclusion and exclusion criteria applied in the PRISMA 2020 methodology. Comment 4: In this paper, the introduction is very long, tiring, with repeated information throughout the text, unconnected sentences, and does not highlight the importance of the work.Response 4: We revised the text accordingly, namely:Climate change, as defined by the Intergovernmental Panel on Climate Change (IPCC), signifies a discernible alteration in the climate's state, identifiable through statistical tests, and persisting over long periods, typically spanning decades or more [1]. This process has garnered consensus among climate researchers, particularly regarding its rapid pace and profound implications. Over the 20th century, Earth's average temperature has surged by approximately 0.8°C, accompanied by an escalation in the frequency and severity of climate-related natural disasters such as cyclonic storms, floods, droughts, and heat waves [1,2].Projections suggest that without active carbon mitigation efforts, global surface temperatures could rise by an additional 2.4-6.4°C by the century's end [2,3]. In light of this, it becomes imperative to understand and address the multifaceted impacts of climate change on various sectors, with agriculture standing prominently among them. Agriculture, being intricately tied to weather and climate patterns, faces manifold challenges arising from alterations in temperature, precipitation, and extreme weather events [2,4].The ramifications of climate change on agriculture are manifold, encompassing shifts in temperature and precipitation patterns, changes in pest and disease dynamics, variations in atmospheric carbon dioxide levels, and modifications in the nutritional quality of crops, among others [5,6].Particularly in Africa, where agriculture serves as a cornerstone of livelihoods, predominantly rainfed and intricately linked to subsistence, the vulnerabilities to climate change are heightened [10,11]. In Africa, where the specter of food insecurity looms large, climate change amplifies existing challenges, potentially jeopardizing efforts to alleviate hunger and malnutrition [10]. Southern Africa, heavily reliant on rainfed agriculture, confronts imminent threats to food security and agricultural productivity due to increased rainfall variability and warming temperatures under climate change [12,13]. Angola, marked by vulnerability to climate change and recurrent droughts epitomize the precarious balance between agricultural sustainability and environmental exigencies [26].The consequences of these shifts extend to staple crops like maize, sorghum, and millet impacting their yields and growing seasons [18]. Millet and sorghum, renowned for their resilience to arid conditions emerge as crucial crops in the face of changing climatic conditions, offering potential avenues for adaptation in regions prone to water stress [19,20]. However, the overarching trend of warming poses challenges to the hydrological cycle, altering precipitation patterns and exacerbating drought conditions [21,22].In light of these considerations, this review delves into the intricate nexus between climate change and agriculture in Angola, exploring the manifold impacts, challenges, and potential avenues for resilience-building in the face of an uncertain climatic future. Through a comprehensive analysis, this study aims to underscore the imperative of proactive measures and policy interventions to safeguard agricultural sustainability and food security amidst the throes of climate change.” Comment 4: Need for updated information on climate scenarios.Response 4: As this is a systematic review, these articles were made available by databases, depending on the search equation and inclusion criteria, and highlight the lack of articles in the region under study, including updated information based on the new SSPs and corresponding climate change projections. Owing to the lack of information on these new scenarios and projections, the authors are already preparing a study in this regard. 

Comment 5: Recent IPCC reports.

Response 5: Thank you for the suggestion. We have added a reference to the latest IPCC Report (AR6). We would like to mention that these reports are not SCOPUS-indexed and thereby not tracked by our search algorithm. Anyway, due to the relevance and pertinence of these reports, we opted to add this reference to our manuscript. Comment 6: References 2 and 3 do not represent the subject.Response 6: The authors thank this comment. Reference 2 indeed provides information on the temperature trend in the introduction, as well as reference 3 in section 2. 

Comment 7: Line 46-50: Repeated words, the sentence will need to be rewritten. Another important point: the increase in the concentration of carbon dioxide is the cause of climate change.... it is necessary to review the way of entering information...

Response 7: We appreciate your suggestion, so we changed the sentence without leaving the central idea we wanted to convey. With the modification made to the introduction, the paragraph underwent some changes (cf. lines 32-35).

Comment 8: Line 51-53: “Among others [5,6]. It is known that agriculture is an economic activity highly dependent on weather and climate to produce the food and fibre necessary to sustain 34 human life [7,8] “: incomplete sentence, with numbering in the middle, requires revision.Response 8: The authors are grateful for the suggestion. With the changes to the introduction, this sentence no longer exists. 

Comment 9: Line53-57: Unnecessary information

Response 9: This information was removed with the reorganization of the introduction. 

Comment 10: Page 5 - 6: The way it was written is not appropriate for a review article.

Response 10: The authors thank this comment. However, this is also a writing trend for many review articles, where the authors raise some questions and their respective hypotheses, answered throughout the work. Thus, we would like to keep this structure.

Comment 11: Line 201-206: This methodology is already known worldwide, there is no need to explain the method. What you need to do is show how the Prisma method was used for this research.

Response 11: Thank you for your feedback and for the opportunity to clarify how we have used the PRISMA methodology. Figure 1 highlights the following steps in the PRISMA methodology to carry out this systematic review.

To improve clarity, we would like to explain in more detail how we have applied the PRISMA methodology:

  1. Review planning: We developed a systematic review protocol that included inclusion and exclusion criteria, search strategies, and study selection methods.
  2. Search for studies: We conducted a systematic search in the Scopus and Web of Science databases, using the search equation in Table 2. We recorded all stages of the search, including the selection criteria used.
  3. Selected studies: We carried out an initial screening of titles and abstracts according to the defined inclusion criteria, as shown in Table 3. We then carried out a complete evaluation of the full texts of the selected studies to determine their relevance.
  4. Data extraction: We developed a data extraction form to collect relevant information from each included study, based on the inclusion and exclusion criteria.
  5. Synthesis of results: We used statistical and narrative methods to synthesize the results of the included studies. This allowed us to identify patterns, gaps, and trends in the reviewed literature, Figures 2 and 3 show this.
  6. Assessment of study quality: We carried out a critical assessment of the methodological quality of the included studies, as recommended by PRISMA.
  7. Preparation of the report: Finally, we followed the PRISMA guidelines to prepare the final report of the systematic review, ensuring transparency and clarity in the presentation of methods and results.

Comment 12: Line 251: repeated information

Response 11: We removed this repetition.

Reviewer 2 Report

Comments and Suggestions for Authors

Title: Is mentioned "impact of climate change . . ." and indicators, however this work is  just a review regarding publications/journals about topics and publications realted. I mean, is difficult to identify imopac of c.limate change in Angola (Africa) and indicators agroclimatics and biocliatics. May be "state of the art of reserachers to analyze impacts of c,limate change in Angola: REVIEW"

Abstract: I think your objective is confuse and ambiguos, as a consequence, at the end of the manuscript I do not qualify if it was reached. Please, to be precise

Introduction; As mentioned in the manuscript comnents, if you are bording indicators and impacts, may be is convenient to explain 

M&M. Authors mentioned some values, ranges, etc, but I do not know, how the authors reach them

Analysis of results and conclusions: In the opinion of this reviewer, these sections are very (very) general, some conditions are applicable in most countries, some considerations are assumptions.

Other specific consideratoins are in the manuscript

Author Response

Response: The authors are grateful to the reviewer for the pertinent suggestions and comments, as well as for the time spent on this revision. Please find our point-by-point replies below (in red font), along with the corresponding changes in the manuscript (also in red font). 

Comments 1: Title: Is mentioned "impact of climate change . . ." and indicators, however this work is  just a review regarding publications/journals about topics and publications realted. I mean, is difficult to identify imopac of c.limate change in Angola (Africa) and indicators agroclimatics and biocliatics. May be "state of the art of reserachers to analyze impacts of c,limate change in Angola: REVIEW"

Response 1: The authors agree on the title of the work, we thank you for the review. In fact, we show some impacts of climate change in Angola resulting from the drought, but as the analysis of agro-climatic and bioclimatic indicators was not carried out, we changed the title to:

“Analysis of the impacts of climate change on agriculture in Angola: Systematic literature review” Comments 2: Abstract: I think your objective is confuse and ambiguos, as a consequence, at the end of the manuscript I do not qualify if it was reached. Please, to be precise.Response 2:Thank you for the suggestion. In order not to leave the objective confusing and ambiguous, we made a point of substantially altering the summary and being as precise as possible.    Comments 2: Introduction; As mentioned in the manuscript comnents, if you are bording indicators and impacts, may be is convenient to explain.

Response 2: Thanks for the suggestion. Due to the change in the theme, no mention was made of the indicators, but we pointed out some impacts of climate change and forecasts for agriculture in Angola, as shown in the manuscript on lines 396-424.

Comments 3: M&M. Authors mentioned some values, ranges, etc, but I do not know, how the authors reach them.

Response 3: The authors thank you for raising this issue related to numbers in M&M. These numbers, ranges, and values were achieved due to the application of the PRISMA2020 methodology. Figure 1 shows these numbers and each category. The search resulted in 431 articles, published between 1993 and 2023.

Comments 3: Analysis of results and conclusions: In the opinion of this reviewer, these sections are very (very) general, some conditions are applicable in most countries, some considerations are assumptions.

Response 3: We agree with this comment. The authors believe that the conclusions are general, with there being an insufficiency in the work regarding the conclusions in the region under study. Therefore, we draw some general conclusions and suggest specific conclusions for the area under study.

Comments 4: Other specific consideratoins are in the manuscript.

Response 4: We agreed with other specific considerations and made a point of responding.

We want to clarify that the paragraphs that appear in the research questions in the review question subsection are the result of the journal template.

Round 2

Reviewer 1 Report

Comments and Suggestions for Authors

Dear Editor,

Some changes were made throughout the text, improving the article. We emphasize the need for a review of the objectives and hypotheses.

The conclusion must be revised to be more objective, removing repetitions of adaptation measures measures. The author must conclude and recommend based on the key objective/aim of the research, a similar thing must be done in the abstract.

What are the adaptation measures for the studied region?

Table 4 is not required.

Author Response

Response: The authors are grateful to the reviewer again for the pertinent suggestions and comments, as well as for the time spent on this revision. Please find our point-by-point replies below (in red font), along with the corresponding changes in the manuscript (also in red font).

The authors would like to thank you for your appreciation regarding the improvement of the article, due to some changes made throughout the text. which is our primary objective.

Comment 1: Some changes were made throughout the text, improving the article. We emphasize the need for a review of the objectives and hypotheses.

Response 1: The authors are grateful for the suggestion to review the objectives and hypotheses. We eliminated question number three and its respective hypothesis and objective since the study did not analyze agroclimatic and bioclimatic indicators. The others remained, respecting the opinion of the other reviewer.

Comment 2: The conclusion must be revised to be more objective, removing repetitions of adaptation measures measures. The author must conclude and recommend based on the key objective/aim of the research, a similar thing must be done in the abstract.

Response 2: We revised some conclusions to be more objective and also eliminated repetitions of adaptation measures. The conclusions were aligned according to the main research objective, we believe that the recommendations will be implicit in adaptation measures and avoid repetitions. In summary, in lines 18-21, we added data on some impacts of climate change in the region under study. We tried to change everything necessary, respecting the suggestions of another reviewer, which were precisely in the summary and the conclusions.

Comment 3: What are the adaptation measures for the studied region?

Response 3: The authors are also grateful for this suggestion regarding adaptation measures in the studied region. We added some adaptation measures for the region.

Line 470-496 “Adapting agriculture to climate change must be a priority to improve crop performance and resilience to environmental constraints. The predicted climate change scenarios, leading to higher temperatures and a water deficit, will impose severe limitations on crop yields and harvested area [36,60].

The Angolan people are vulnerable during prolonged periods of drought: a total of 11.1 million people (37%) of the population live in rural regions, and the majority of them practice rain-dependent subsistence agriculture [15]. Therefore, there is a need to adopt adaptation measures to mitigate the impacts of climate change on agriculture.

For the southern region specifically, the adaptation strategies suggested for crops and cultivation systems include the genetic improvement of corn, and the development of new cultivars with high productive potential and better adaptation to water deficit conditions and high temperatures, which are: millet (Pennisetum glaucum) and sorghum (Sorghum bicolor), in the case of flooding, rice is the only main cereal adapted to flooding.

As for the cultivation system, we suggest adjusting the cultivation calendar to avoid the hottest periods of the year in order to help reduce thermal stress on plants.

Regarding the agricultural system, it is suggested investing in sustainable irrigation systems, such as drip systems and rainwater capture, to guarantee the supply of water to crops during periods of drought and introducing sustainable agricultural techniques, such as conservation and the use of cover crops, to improve soil health and increase crop resilience to extreme weather conditions.

Finally, provide training and education to farmers on sustainable agricultural practices, water management and climate change adaptation techniques and have early warning systems in place to help communities prepare for extreme weather events such as droughts and floods, and reduce the damage caused by these events.

Comment 4: Table 4 is not required.Response 4: We appreciate your suggestion regarding Table 4, but the inclusion of this table provides transparency and methodological accuracy to the systematic review, allowing readers to fully understand the research process conducted. Additionally, the table serves as a valuable reference for other researchers wishing to replicate or expand the study. 
